# Polymeric Materials Used for Immobilisation of Bacteria for the Bioremediation of Contaminants in Water

**DOI:** 10.3390/polym13071073

**Published:** 2021-03-29

**Authors:** Dmitriy Berillo, Areej Al-Jwaid, Jonathan Caplin

**Affiliations:** 1School of Pharmacy and Biomolecular Sciences, University of Brighton, Brighton BN2 4GJ, UK; 2Department of Biotechnology, Al-Farabi Kazakh National University, Almaty 050040, Kazakhstan; 3Department of Pharmaceutical and Toxicological Chemistry, Pharmacognosy and Botany School of Pharmacy, Asfendiyarov Kazakh National Medical University, Almaty 050000, Kazakhstan; 4School of Environment and Technology, University of Brighton, Brighton BN2 4GJ, UK; Areej.Khudhair@stu.edu.iq (A.A.-J.); jonlsc@outlook.com (J.C.); 5Environment and Pollution Engineering Technical Department, Basrah Engineering Technical College, Southern Technical University, Basra 61003, Iraq

**Keywords:** cryogels, bacteria immobilisation, bioremediation, water purification, biofilm

## Abstract

Bioremediation is a key process for reclaiming polluted soil and water by the use of biological agents. A commonly used approach aims to neutralise or remove harmful pollutants from contaminated areas using live microorganisms. Generally, immobilised microorganisms rather than planktonic cells have been used in bioremediation methods. Activated carbon, inorganic minerals (clays, metal oxides, zeolites), and agricultural waste products are acceptable substrates for the immobilisation of bacteria, although there are limitations with biomass loading and the issue with leaching of bacteria during the process. Various synthetic and natural polymers with different functional groups have been used successfully for the efficient immobilisation of microorganisms and cells. Promise has been shown using macroporous materials including cryogels with entrapped bacteria or cells in applications for water treatment and biotechnology. A cryogel is a macroporous polymeric gel formed at sub-zero temperatures through a process known as cryogelation. Macroporous hydrogels have been used to make scaffolds or supports for immobilising bacterial, viral, and other cells. The production of composite materials with immobilised cells possessing suitable mechanical and chemical stability, porosity, elasticity, and biocompatibility suggests that these materials are potential candidates for a range of applications within applied microbiology, biotechnology, and research. This review evaluates applications of macroporous cryogels as tools for the bioremediation of contaminants in wastewater.

## 1. Introduction

Current multistep water purification methods to remove chemical contaminants include hydrogenation, ion exchange, liquid–liquid extraction, adsorption by activated carbon, forward and inverse osmosis, electrolysis, sonochemistry, UV irradiation, and oxidation. Some processes require high temperatures and extreme pressures (e.g., hydrogenation, hydrodechlorination), and/or expensive catalysts such as platinum, palladium, rhodium, and gold nanoparticles on carbon or other supports [1,2,3,4,5], and are therefore not always cost-effective. Some compounds may result in the formation of more toxic or mutagenic derivatives. For example, the Fenton oxidation process, which usually involves the use of hydrogen peroxide and salts of iron (II) and generating free highly active hydroxide radicals may result in the production of polychlorinated dibenzo-p-dioxins and dibenzofurans [3,4,6]. The chemical redox process using iron nanoparticles (FeNPs) for the dechlorination process, where the dissolution of iron in water leads to the generation of radicals of hydrogen that, in turn, are involved in the interaction with organic molecules, and the other mechanism may include the interaction of iron oxide/hydroxide shell of FeNPs with chlorinated functional groups [5]. Several book chapters are devoted to immobilisation strategies for microorganisms (prokaryotes and eukaryotes, primarily fungi) on various organic (bark or wood chips) and inorganic supports (Fe_2_O_3_, TiO_2_, Al_2_O_3_ x SiO_2_, cement particles) [5,7]. Bertrand et al. made a comprehensive overview of a range of efficient systems of waste water treatment at the industrial scale and accompanied each case study with a simplified yet very informative illustration of the principles of the system [5]. Advantages and drawbacks of each immobilisation approach (artificially generated biofilm, biofilm produced in nature) compared to microorganisms in a planktonic state for purifying of waste water from oil, herbicides, pesticides, xenobiotics, and heavy metals are discussed [7]. These approaches for water treatment are quite efficient but at the same time expensive and in most cases quite specific to the contaminant. Such methods require complex technological processes, which are expensive to build and operate. In comparison, biological treatment methods have fewer environmental impacts and are more energy-efficient, but their performance is limited at high concentrations of contaminants. Enzymatic treatment protocols are immobilized to be an efficient way to treat various compounds. However, they are unstable in the free state and require immobilisation on a substrate to ensure durability and efficacy. To reduce costs, whole cells can be used as the enzymatic system. The immobilisation of microorganisms onto appropriate substrates can be exploited for a number of processes and allows easy recovery and reuse of the bacteria [8]. Immobilisation can enhance the operational stability of cells and can protect them from the effects of extreme pH, toxic compounds, turbulent reaction technologies, and reduce the risk of contamination of cell cultures [4,9,10,11]. The review by Martines et al. [10] was devoted to a brief chemical description of each natural (Chitosan(CHI), Alginate(Alg), agar, collagen, agarose) and synthetic polymers (poly-acrylamide(pAAm), polyvinyl alcohol (PVA), polyethylene-glycol (PEG) and polycarbamoyl sulphonate (PES), polypropylene (PP), polyethylene (PE), polyvinylchloride (PVC), poly-urethane (PU), polyacrylonitrile (PAN)) that are utilised for the preparation of inexpensive, non-toxic, and potentially with reactive functional groups carriers for entrapment of microorganisms.

Bacteria have been immobilised on a range of chemically activated and/or inert supports. These include biofilm formation on activated carbon, powdered activated carbon with external surface areas of 938–978 mg/m^2^, inorganic metal oxides, membranes or porous polymers, and chitosan-beads [12,13]. Ionic liquids have been used in sensor-based applications as they significantly prolong the lifetime of the sensor and enable the number of immobilised cells on the electrode surface to be controlled [14]. The management of bioremediation process parameters such as oxygen concentration, pH, sulphide, and nitrite/nitrate levels is important, especially processes utilising entrapped bacterial cells. The bacterial consortium immobilised on natural coconut coir is illustrated to be the most efficient matrix to enhance the effectiveness of sewage remediation. Reactors operated in continuous mode for 15 and 16 days revealed 75.9% and 73.7% chemical oxygen demand decline and removal of nitrogen NO_3_ of 78.9% and NH_4_ of 79.7%, respectively. The organic contaminant removal rate was not significant during the first day; however, from the second day, the removal efficacy was continuously increased, due to adaptation of the bacterial community, which takes 24 h. On the eighth day, a fifty per cent decline of contaminant was registered. The maximum kinetics of remediation was exhibited on the 15th day in domestic wastewater column reactor experiments [15]. The density of bacteria on the substrate material depends on the structure, pore size, and surface area of the support, as well as the nature of the material (hydrophobicity, charge, etc.) and environmental conditions, such as ionic strength, presence of some trace essential elements, pH, and temperature. It has been discussed that the bioremediation rate can be enhanced using immobilisation methods, particularly under harsh conditions [16]. Immobilised cells are easier to separate, are more tolerant to pH and temperature changes, and have possible reutilisation potential [17,18]. Efremenko et al. studied the effect of cell storage at −18 °C for 18–24 months on reproductive capacity of different microorganisms (Gram-positive and Gram-negative bacteria, yeasts, and filamentous fungi) entrapped in PVA cryogel. It was observed that activity of immobilised *Escherichia coli* cells preserved in granules of PVA cryogel revealed a high level of productivity of a target recombinant protein over 1.5 years [19]. Contamination of the environment with AAm is great, as it is a side product of various processes besides enormous industrial production of polyacrylamide. AAm degrades quickly at low concentrations (10 mg/L) in surface water; however, at higher concentrations, it stays stable for approximately 60 days in tap water. Such a problem for wastewater can be solved by bioremediation of AAm solution (1–5%) by *Pseudomonas aeruginosa* immobilised into Ca-Alg hydrogel within 2–3 days and faster than free bacteria. Moreover, the presence of 200 and 400 mg/L of nickel enhanced the process [20].

Cortez and co-workers presented a review of adhesive biocatalytic coatings (biocoatings), with a nanoporous microstructure of polymeric particles incorporated with a high density of viable cells [18]. Lyophilised biocoatings stabilised with carbohydrate osmo-protectants can be prepared by high-speed coating technologies, aerosol delivery, or ink-jet printers. They form multi-layered, patterned coatings on flexible nonporous or nonwoven substrates, preserving the order of 10^10^–10^12^ non-growing viable bacteria per m^2^ [19]. The structure of the biocoatings allows illumination of a high concentration of photo-reactive cells or algae within thin liquid films for efficient mass transfer. Biofilm formation, however, is usually a long process and represents the main cost and limitations of biotechnological processes using immobilised bacteria (Figure 1b). Biofilm develops in the following steps: motile cells attach to the surface, maturate, and produce protective extracellular matrix, finally followed by biofilm dispersal [21,22]. Thermosensetive cryogels composed of co-polymer N-isopropylacrylamide with N-hydroxyethyl-acrylamide was modified using controlled polymerisation by glycomonomer (mannose, glucose). These cryogels were proposed for sample preparation of water after purification at the quality control stage for accumulation of potentially photogenic bacteria at low concentrations via filtration of a large volume of water, followed by elution by sugar-containing eluent [22]. This approach also can be utilised for accelerated biofilm formation for various biotechnological processes.

## 2. Application of Immobilised Bacteria

Different cryogel configurations have been developed to immobilise bacteria, including entrapment in a polymeric matrix or attachment to a solid support. Three main methods of immobilisation are illustrated in Figure 1. Entrapment and encapsulation of microorganism into the polymer network with restricted diffusion of substrate to cells and corresponding metabolism products, low density of cells relatively hydrated polymer volume ratio, high viability of cells is illustrated onFigure 1a [24,25].

Figure 1b shows biofilm generation on the porous surface of inorganic supports (silicates, perlite, vermiculite, Grow plant or glass aerogel) [26] or scaffold such as composite cryogels [27]. Microorganisms in a biofilm form are robust and usually show tolerance to toxic ions that might be present in the contaminated water due to cells incorporated in the biofilm matrix, which acts as a physical barrier. These advantages make biofilm attractive viable catalysts for organic syntheses, which may find wide technological applications [7]. In Figure 1c, macroporous hydrogel formation takes place via direct covalent linkage of bacteria cells’ membranes’ into 3D biofilm(advantages: one step, rapid biofilm formation due to cryoconcentration phenomenon leading to generation of high cells density with respect to polymer weight) [23,28,29]. *Pseudomonas fluorescens(S3X)*, *Microbacterium oxydans (EC29)* and *Cupriavidus* spp. (1C2) were immobilised on the hydroxyapatite surface obtained from fish’s bones with a surface area in the range of 15.5–24.4 m^2^/g within one hour at 30 °C at agitation. This approach illustrated a concurrent effect of adsorbed bacteria and the material resulting in a synergistic effect of improved removal efficiencies of zinc and cadmium ions from solution [26].

A microbial fuel cell (MFC) is a closed fuel cell system composed of an anode and cathode, where bacterial cell-mediated electron transfer takes place from the oxidation of an electron donor compound and reduction of electron acceptor functionality. The main application of MFC is to treat wastes for electricity generation [7]. A microbial electrolysis cell (MEC) is a bioelectrochemical system containing an anode and cathode, where anodic microorganisms oxidise organic derivative and the resulting protons are reduced to hydrogen atoms, forming gaseous hydrogen molecules. The reaction on the cathode electrode may also result in the formation of methane, which depends on immobilised bacteria [7].

The different methods for immobilisation will affect the performance of the cells and may also impact the number of viable cells and hence the performance of the biocatalytic or bioremediation process [27]. However, the number of viable cells in these experiments was not quantitatively estimated via biochemical assays, and therefore, it is difficult to obtain reproducible results. Nevertheless, immobilisation strategies illustrate benefits compared to planktonic free cells, leading to significantly improved remediation rates [30,31,32]. The principal difference between cryogels and other porous materials(perlite, vermiculite, Grow plant or glass aerogel) of similar pore size is that the former exhibit tissue-like elasticity and flexibility and are able to withstand deformation without being destroyed or damaged. Preparation at low temperatures allows sufficient preservation of enzymes or cells (Figure 1c) [23,28,29].

Cryogels possessing a spongy 3D polymeric network structure are synthesised via a cryogelation or cryostructuration technique [30,31,32,33,34]; the concept of cryogel preparation based on cells is summarised in Figure 2 [29]. The available cryogelation techniques enable the production of elastic macroporous cryogels with a wide range of porosities and morphologies; for instance, Figure 3 exhibits the morphology of composite cryogels based on PVA, chitosan, and glutaraldehyde(GA) and PVA, CHI, hydroxyapatite, and GA [35]. The change of cryogelation mode gives the possibility to regulate material mechanical and physico-chemical properties (porosity, crystallinity, melting point, etc.) [5,29,31]. Cryogels can be prepared by applying several approaches, such as free-radical polymerization of a monomeric mixture or suspension with particles (metals, polymers, cells, bacteria) [25,36,37,38,39]. Other approaches include irradiation of the polymer solution or suspension with gamma-rays (electron beam, UV) [40] or modified polymers and self-assembly of supramolecular gelators [5,39,41]. The morphological characteristics of macroporous cryogels produced by cryogelation are controlled by parameters such as the rate of crystallisation of the solvent and the size of the solvent crystals, type of the solvent (water, DMSO, dioxan) [29,31,42,43], and the phase-separation between the polymers and crystalizing solvent. For the process of hydrogel formation, covalent cross-linkers such as N,N-methylenbisacrylamide, PEG-diacrylate, glutaraldehyde(GA), oxidised dextran(Ox.Dex.), and tetramethoxypropane have been tried [28,44,45]. Noncovalent cross-linking methods include noble metal complexes, hydrophobic interactions, π-πstacking, polyelectrolyte complexes, and hydrogen bonding [33,39,44,45]. Furthermore, the temperature and the rate of freezing can impact the formation and growth of ice nuclei and hence pore size. If a solution is frozen in liquid nitrogen (−196 °C) it results in the rapid formation of ice nuclei and the growth of small ice crystals, leading to poor water permeability. If the freezing takes place at a higher temperature (e.g.,−20 to 8 °C), ice nucleation is slower and the nuclei tend to grow into larger ice crystals, which results in the preparation of materials with large and randomly oriented pores after the freeze-drying process. The direction of freezing has a major effect on pore morphology; thus, by controlling the direction of freezing, the growth of ice crystal can be orientated in one direction due to a great temperature gradient across the sample with the ice crystals growing from the low to the higher temperature end [46]. Low temperatures (−10 to −25 °C) in water-based solvent systems enable the development of large solvent crystals and thus pore sizes [27]. Control of the pore size of the cryogel is achieved using a pre-freezing method, which effectively controls the freezing of large volumes of solvent and polymer solutions. The mixture is frozen to below the freezing point of the solvent with constant stirring prior to initiating gelation, leading to the freezing of up to 90% of the solvent, whilst the gel-forming compounds remain in a non-frozen liquid state. The agitation of the reaction mix improves the heat transfer and formation of ice nuclei. Cryogels formed this way have large pore volumes and near-uniform porosity with interconnected pores of diameter 20–200 µm, and exhibit twice the strength of cryogels produced by conventional methods (Young’s modulus = 12 kPa vs. 6 kPa) [43]. Covalent attachment of cells to the support is one of the most widely utilised methods that links the micro-organism with bonding reaction of reactive functionality (amino (–NH2); hydroxyl(–OH)thiol(–SH) or carboxygroups(–COOH)) at the cell’s surface. This leads to the stability of the microorganism, but the bioactivity of bacteria might be slowed down. Usually, inactive carbonyl groups presented in Alg, gelatin, and carboxymethylcellulose requires prior activation with carbodiimide for the purpose of immobilisation. Hydroxyl groups on the surface of metal oxides can be functionalised using various silane derivatives (aminopropyl -triethoxysilane, 3-(Trimethoxysilyl)propyl methacrylate, etc.) [47] Polymers containing amino(–NH2, –NH–), hydroxyl (>C–OH), and thiol(>C–SH)can be functionalised with an active epoxy group via treatment with epichlorohydrine or di- and triepoxyderivatives [32].

The physical entrapment of bacteria in PVA cryogels was one of the first applications in microbiology [34,48] (Figure 1a), where the viability of bacteria within the cryogel and effect of temperature stress were estimated via bioluminescent activity of immobilised photobacteria [49]. A production of lactic, fumaric and succinic acids by various microbial cells (filamentous fungi *Rhizopusoryzae* (F-814, F-1127) and bacteria *Actinobacillus succinogenes* B-10111) immobilised into PVA cryogels revealed higher yields than non-immobilised cells [50]. The macroporous cryogels revealed enhanced efficacy with various cell types compared to polymeric matrices available at the time such as pAAm, Alg, hydroxymethylmethacrylate (HEMA), humic acid particles cross-linked by triglycidylpropane derivative within gum Arabic, and carrageenan [51,52,53,54]. Currently used carriers can be classified as organic carriers, inorganic carriers, and composite carriers regarding their chemical composition. Organic carriers are subdivided into natural polymeric carriers and synthetic polymers [51]. Hydrophobicity, charge of the surface, roughness, morphology and texture of carriers, flow rate, and temperature of the process are factors that affect microbial cell adsorption and therefore have to be taken into account for efficient cell immobilisation. The review contains methods of immobilisation of *Chlorella* spp. into Alg, carrageenan agarose, Alg and agar beads, polyurethane foam, and AAm gels for the removal of harmful heavy metals (Ni, Zn, Cd, Cu, Hg, Pb, and uranium). Moreover, those bioremediation systems illustrated 4–5 times more efficiency toward phosphate, nitrate, and ammonia removal in comparison with free cells [51]. There are quite a few publications related to pilot-scale studies. A recent report ascribes a novel design of immobilization of *Pseudomonas citronellolis* on a composite of PVA bamboo-biochar beads (cell-biochar beads), which is environmentally friendly scaffold that was utilised for biotrickling filters (BTFs) fabrication. On one hand, cell-biochar beads provide adsorption of pollutants. On the other hand, it works as a carrier for the attachment and proliferation of bacteria. Therefore, BTFs illustrated enhanced efficiency of pollutant removal (toluene removal was over 99%). The organic loading rate and elimination capacity of BTF for toluene were 46 g/m^3^∙h and 35 g/m^3^∙h at a gas flow rate of 0.015 m^3^/min, respectively. Cell-immobilised biochar beads were made by mixing 50 g of 9% PVA solution, 10 mL of bacterial strain, and 65 g of bamboo-bio-char powder (grain size 0.15 mm) [52]. Aslıyüce et al. [54] studied the efficiency of cryogels produced via direct radical polymerisation of (hydroxyethylmethacrylate)HEMA with freeze-dried bacteria *Trichoderma* sp. for the removal of cyanide (20ppm) at a wide pH range of 4–8 and temperature range of 20–40 °C. Berillo et al. [29] tried to reproduce the experiment with direct polymerisation of different bacteria strains. Metabolic activity Cell Viability MTT assays showed that cell viability was less than 5%, most probably due to damage to the cell membrane by active radicals. Stepanov et al. [55] demonstrated the successful entrapment of *Komagataei bacterxylinum* B-12429 into a PVA cryogel. The bacteria produced cellulose, which passed through the pores of cryogel matrix and accumulated on the surface of the medium. The synthesis of bacterial cellulose was 1.3–1.8-fold higher than the planktonic bioreactor. Repeated utilization of the immobilised bacteria preserved 100% of their metabolic activity for 10 working cycles (60 days). Some researchers state that *Bacillus pseudomycoides* immobilized PVA)/GA hydrogel at mass ratio of 0.03 and acidic pH of 2 maintaining cell accessibility to external environment for bioremediation of wastewater, [56] however it won’t work for most of low pH non-resistant bacteria strains. Recent evaluation of patent applications revealed most efficient series of bioreactor design (magnetic fluidised bed reactors, fluidised bed bioreactor employing biofilm carriers, reverse fluidised loop reactor, fluidised bed reactor, packed-bed reactor, biological permeable barrier for removal groundwater contaminations, and layered packed bed bioreactor) [47].

Although cryogels have been in development since the mid-1980s, it is only within the last few years that the potential of macroporous cryogels for cell immobilisation in cell culture applications and in bioremediation approaches has been realised [19,34,35,38,56,57,58,59,60]. Lysozyme-imprinted bacterial cellulose (Lyz-MIP/BC) nanofibers were developed to recognise lysozyme via metal ion coordination. This material showed high selectivity for lysozyme towards bovine serum albumin and cytochrome c adsorption as a proof of concept for purification of biological fluids from complex target molecules [37].

The inherent characteristics of macroporous cryogels, such as their high flow permeability, low flow resistance, and minimal toxicity [60], make them suitable for applications in a range of processes including aerobic and anaerobic bioreactors [28,61,62,63,64], the separation of cells and bio-particles [27,56,65,66,67], scaffolds for tissue engineering and biomedical applications [35,59,68], and for the perfusion of blood products [69,70,71].

## 3. Design of Functional Macroporous Cryogels

Composite cryogels comprising the strong anion-exchange resin Amberlite IRA-401 or other functional monomer iminodiacetic acid moiety, mimicking Immobilised Metal Affinity Chromatography (IMAC) columns have been evaluated [27,38]. Monolithic columns with pores of up to 100 µm in diameter with a sponge-like morphology were trialled as adsorption media for the bio-separation of particulates from fluids and the efficient separation of whole cells from media, based on electrostatic interactions and a cooperative effect [27,38]. The efficiency of IMAC cryogels was assessed using mixtures of wild-type *Escherichia coli* (w.t. *E. coli*) and recombinant *E. coli* cells displaying poly-His peptides (His-tagged *E. coli*), and also of w.t. *E. coli* and *Bacillus halodurans*. Wild-type *E. coli* and His-tagged *E. coli* were quantitatively separated from the media containing equal amounts of both cell types. The cells were recovered by selective elution with imidazole and EDTA and showed yields of 80% and 77%, respectively [59,65,67].

A modification involves forming the macroporous cryogel within a protective polymeric core to render the cryogels resistant to shear forces encountered during stirring in a bioreactor [72]. The cores, known as Kaldnes carriers [73], were originally used as supports for bacterial biofilms for utilisation in moving-bed biofilm reactors (MBBR) employed for biological wastewater treatment. The polymeric cores reinforced the mechanical stability of cryogels used for bioremediation or biotechnological process, such as biogas production [31,74,75,76,77] (Figure 4).

The carriers are designed to provide a large protected surface area for the microbial biofilm to grow and optimal conditions for the bacteria culture when the carriers are suspended in water. A brief classification of carriers by chemical composition used for immobilisation of microorganisms is presented in Figure 4. Moreover, the exploitation of cryogels prepared within polymeric carriers is applicable to ordinary water treatment approaches such as activated sludge or immobilised bacteria on a high density support of silica, titanium, and alumina oxides [32]. The bioremediation of oil spillages in the marine environment is predominantly a surface water phenomenon, and the application of low density, very high porosity(Figure 4), and floating materials has been found to be effective [29]. In the biomedical and biotechnological fields, bulk macroporous hydrogels with defined three-dimensional structures and large surface areas and volumes, with high permeability and low flow resistance, are desired. These factors are dependent on the techniques and materials utilised to form the cryogel [30,67]. Methods commonly used result in the production of cryogels of low volume and size (diameter 5–10 mm, length 20–40 mm), which limits their application. However, Cunningham et al. developed a novel method for the synthesis of large-volume, [43] three-dimensional macroporous gels with large surface area and wide interconnected pores of sizes in the range 50 to 200 µm in diameter. The approach produced cryogels of 400 mL volume with controlled pore structure and size.

## 4. Microorganisms Immobilisation Strategies into Cryogel Structure

The addition of cell-specific ligands to the cryogel enables the specific binding of viruses [66]. A streptavidin modified cryogel was designed to enable single-step capture of chemically biotinylated Moloney Murine Leukaemia Virus from crude, unclarified cell culture supernatant [66]. Scanning electron microscopy indicated that the bound cells were distributed uniformly throughout the mesh of pores without physical entrapment of the cells (Figure 1c) [23]. The high density of the cell suspensions did not block the pores of the cryogel, thus not impeding the flow permeability rates [19,27]. In applications of macroporous cryogels containing bacteria, the bacterial cells are either encapsulated within the pore walls, adsorbed to the cryogel surfaces, or covalently bound to the polymer walls (Figure 1a,b) [30]. An alternative approach developed by Kirsebom et al. [78] is to form cryogels from the actual bacteria by crosslinking the bacterial cells using glutaraldehyde. A drawback is that GA may adversely affect the growth and viability of the bacterial cells since the GA molecule is small enough to enter the cell and disrupt cellular metabolism [23,28,79]. Consequently, the concentration of GA (and other low-molecular-weight aldehydes employed as cross-linkers) requires careful calibration to achieve a balance between cryogel morphology and strength and the viability of the bacterial cells. One of the first attempts to use GA as a cross-linking agent for whole-bacteria immobilisation was tested on *Trigonopsis variabilis* CBS 4095, which was treated with GA, and polyethyleneimine (PEI) and as a result, a specific activity of D-amino acid oxidase was 82–98 U/g dry biomass, which corresponded to about 20% of untreated cells [80]. A similar tendency was observed when GA was utilized as a cross-linker under cryoconditions for *E. coli*. The b-glucosidase activity of cryogel from *E. coli* was only 15% for b-glucosidase-induced cells [28]. It has been revealed that high-molecular-weight aldehydes are less damaging to microorganisms [28,62,79], as are more biocompatible cross-linkers such as Ox.Dex., with molecular weight in the range of 40–500 kDa [28,62,81].

The polymeric cryogels containing Ox.Dex. exhibited satisfactory mechanical properties just after thawing of the cryogel (Young’s modulus up to 2.8 kPa) showed strong biodegradability and biocompatibility. Dextran derivatives were not efficient for cross-linking bacterial cells due to the lack of formation of Schiff’s base at the temperatures used and the sensitivity of Schiff’s bases to the acidic pH of the medium [45]. Rapid decomposition was also observed in a control cryogel without enzymatic activity composed of Ox.Dex. with gelatin [45]. Cryogels with typical porosity just after preparation and hydrolysis led to unlimited swelling and poor water permeability (0.04 mL/min) but satisfactory mechanical properties (1.85 kPa) [28]. N evertheless, the Schiff’s base groups did form in this system under moderate heating (60 °C) in the dry state, probably due to the formation of semiacetal groups within the dextran structure [82]. This drawback seen with some modified natural polymers illustrates the necessity to apply synthetic polymers that are more resilient to degradation. A very recent report illustrates the good biocompatibility of polypyrrole-gelatin cryogels for culturing the fibroblast cell line (NIH/3T3). Furthermore, this scaffold showed weak antibacterial activity against *Staphylococcus aureus* (CCM 4516) and *E. coli* (CCM 4517), according the ČSN EN ISO 20743: 2014 [27]. In an attempt to mitigate damage to the bacterial cells during cryogelation, when GA was applied as a cross-linker, Börner et al. [62] developed and utilised milder polymer-based cross-linkers such as Ox.Dex., and a mixture of aldehyde-modified poly(vinyl alcohol) (PVA-al) and aldehyde activated polyethyleneimine (PEI-al), resulting in a cryogel containing cross-linked cells of *Clostridium acetobutylicum* DSM 792 for the production of butanol. Due to the significantly lower toxicity of the polymers, acetone-butanol-ethanol production was improved 2.7-fold in immobilised cells in comparison to free cells. The ability to recycle the cryogels 3–5 times with further aliquots of contaminated water was exhibited. This was a proof of concept confirming the possibility of the utilisation of cryogels under anaerobic conditions, which may find application in anaerobic wastewater treatment using related bacterial strains. The resultant solid, elastic cryogel had large porosity with good rates of mass transfer in the butanol production process. Zaushitsyna et al. [28] optimised conditions of utilisation of novel polymers PVA-al and PEI-al in terms of elasticity and water permeability using *E. coli* strain with b-Glucosidase activity. It was found that among tested compositions, the use of final concentration of PEI-al and PVA-al at 0.55% and 0.35%, respectively, resulted in self-supporting scaffolds with a Young’s modulus of 3.15 ± 0.39kPa and water permeability of 0.47 ± 0.17 mL/min, as well as illustrating the best enzymatic activity.

To evaluate the changes in mechanical properties and the degradation of porous material made from viable *Pseudomonas* spp., *Rhodococcus* spp., and a 50:50 mixture of both more accurately, the authors measured the rheological parameters before and after the bioremediation process [31]. FTIR spectra of the activated PEI and PVA revealed peaks representative of carbonyl groups at 1717 and 1714 cm^−1^, suggesting that free aldehydes were present in the polymer [28]. These functional groups would be available for reaction with lysine residues on the membrane of the bacterial cell wall, enabling multiple attachment sites on the cell wall without penetration of the cell wall itself [81]. The combination of aldehyde-activated PVA and PEI resulted in 90% β-glucoronidase activity in cross-linked *E. coli* cells, whereas *E. coli* cross-linked with GA alone exhibited a complete loss of activity [28].

## 5. Applications of Macroporous Cryogels for the Remediation of Contaminants in Water

A number of applications utilising cryogels have been described for the removal of contaminants during the water treatment process. These include wastewater purification and the removal of endocrine-disrupting compounds [83], and the removal of oils and phenols in petrochemical and wood processing wastes [10,27,47,58,82,83,84] Mattiasson al. have developed a highly effective enzyme or bacteria-free molecular imprinted cryogel for removal of endocrine-disrupting compounds (17b-estradiol(EP), 4-nonylphenol, atrazine) from aquatic environments. Such efficacy was achieved due to the utilisation of the molecular imprinting approach, which is quite expensive, but is robust to degradation and allows to be used for a number of cycles. The PVA cryogel with molecularly imprinted polymer particles(MIP) of 4-vinylpyridine, ethylene glycol dimethacrylate, methacrylic acid, and ethylaminoethyl methacrylate were synthesised and adsorption capacity studied [85]. In moving-bed reactor mode, 100% removal of 17beta-estradiol (E2) was observed after applying of E2 solution (10 L, 0.5 ug/L) to the column filled with the E2-MIP/Macroporous Gel Particles at a flow rate of 15 mL/min, whereas 77% of the E2 was removed from the applied solution by the non-imprinted polymer [85]. Atrazine removal efficiency for MIP and non-imprinted polymer were 75% and 26%, with a contact time of 8 min after applying 5 L of a 10 ug/L atrazine solution at room temperature, respectively. The removal of atrazine from the aqueous solution was incomplete, and the final concentration was 1.22 ug/L, which is lower than the current concentration of atrazine in drinking water (3 ug/L). Utilisation of molecular imprinting technology is applicable when highly hydrophilic and toxic compounds should be removed from drinking water, if treatment with activated carbon has low efficiency.

See at al. studied the system of PVA-cryogel-immobilised biomass with 0.5% (*w*/*v*) powdered activated carbon. It was noted that both the immobilised and planktonic mixed cell cultures reached the same removal efficacy of o-cresol (300 mg/L) degradation. However, the ability of free biomass to o-cresol(400 mg/L) degradation decreased significantly compared to the immobilised PVA cryogel. PVA cryogel in 30 h of incubation resulted in 100% removal, whereas the suspension of biomass reached only 45% [86]. The role of microbes in the removal of toxic pollutants from waste water involves a large-scale sophisticated biodegradation process. Natural bioremediation plays an important role in modulating the microbial activity, which has been illustrated to be influenced by various environmental factors and is therefore slow for some contaminants. A comprehensive review of various *Pseudomonas* sp. immobilised on polymeric support was ascribed for the bioremediation of environmental pollutants [58]. It is known that the biodegradation rate diminishes with the number of halogen atoms in the derivative: two halogen atoms in ortho position on the same aromatic ring (2,6) or on different rings (2,2′) are biodegradation-resistant; moreover, an aromatic ring cleavage predominantly occurs on an unsubstituted ring [5]. A combination of anoxic and aerobic treatments is needed for effective mineralization of polychlorinated compounds; for example, after 4 days, bioremediation of most of the mono- and about 25% of the dichlorobiphenyls were degraded, and major steps of aerobic biodegradation pathways are well illustrated [5]. The bioremediation of stable and toxic pollutants has a negative effect on the activity of the biomass used in the process. In one approach to overcome, activated sludge was entrapped into (poly)vinylalcohol (PVA) cryogel beads. Their external surfaces were coated with powdered activated carbon for protection against chlorophenols. Contaminant concentrations were reduced to optimum levels, and the biomass was stable for over 20 cycles [87,88]. Whilst initial adsorption was rapid, it took up to 24 h to reach the adsorption equilibrium, and adsorption was maximal under acidic conditions. Despite these drawbacks, the porous carbon amorphous structure had a low surface area (0.5–1.5 m^2^/g) and mesoporosity, which are considered ideal characteristics for the adsorption of pesticides, oil, herbicides, metabolites of drugs, etc. It was shown that a wide range of organic compounds, e.g.,phenol, nitrophenol, resorcinol, dioxane, acetonitrile and benzotriazole (polar), and toluene, styrene, cyclohexane, paraxylene, benzene, and hexane (nonpolar), can inhibit the bioremediation process in activated sludge. It was summarised that eleven out of thirteen compounds significantly inhibited the activated sludge activity. Polar compounds had a more pronounced inhibitory effect, while non-polar hydrocarbons had relatively less effect on inhibition of an activated sludge activity at high concentrations [87,88].

A recent review by Alessandrello et al. discusses achievements in the usage of various PVA-entrapped bacteria strains (*Rhodococcus* spp., *Bacillus* spp., *Pseudomonas taiwanensis*, *Acinetobacter baumannii*, *Rhodococcus opacus* and *Rhodococcus ruber*) [89].Other immobilisation techniques for the efficient remediation of water from polyaromatic hydrocarbons and oil-polluted waters were investigated by Serebrennikova et al. [90] A reusable fluorescent ion-sensing cryogels polyacrylamide (PAAm-Pyr) obtained via the post functionalisation of macroporous PAAm-Pyrgel with a pyrene moiety was developed for the detection of extremely low concentrations (2 ppb) of mercury (Hg^2+^) in aqueous solutions. This material possessed high selectivity for Hg^2+^ in the occurrence of competitive ions in water [91].

For the treatment of microbially contaminated water, approaches utilising the inherent properties of antimicrobial peptides (AMPs) have been employed. AMPs are natural and synthetic oligopeptides thatare effective against viruses, bacteria, fungi, and parasites [92]. In contrast to conventional antibiotics, some AMPs can reduce the inflammatory response by blocking lipopolysaccharide (LPS)-induced cytokine release by macrophages, which is a major cause of sepsis during antibiotic therapy [93]. Sahiner et al. describe a design and application of Gum Arabic linked with divinyl sulfone cryogel and degradable and biocompatible humic acid nanoparticles crosslinked by trimethylolpropane triglycidyl ether, poly(ethylene glycol) diglycidyl ether, and trisodium trimetaphosphate [46]. These composite cryogels with silver or copper nanoparticles possess antimicrobial activity against *Escherichia coli*, *Staphylococcus aureus,* and *Bacillus subtilis* and can be considered for the final stage of water treatment [46]. Shirbin et al. [94] developed a synthetic oligopeptide-based macroporous cryogel for water purification using a polylysine-b-polyvaline polymer linked by GA. The cryogel was capable of a 95.6% reduction in viable *E. coli* cells after one-hour incubation, whilst a control hydrogel without the AMP showed a less than 20% reduction in *E. coli*. The authors suggest that the increased antimicrobial surface area of the pores, together with their macroporous network structure, results in a “trap and kill” mechanism [94].

The potential toxicity of crosslinking polymers is a limitation for their use in cryogel production. The acrylamide monomer is regulated by the European Union’s Registration, Evaluation, Authorisation and Restriction of Chemicals list due to its potential mutagenic, teratogenic, and carcinogenic properties and is included in the list of substances of concern [95]. As a result, workers have turned to other less toxic compounds.

Within the author’s lab aldehyde activated poly(vinyl alcohol)—PVA-al and polyethyleneimine—PEI-al cryogels cross-linked with different phenol-degrading bacteria strains have been developed to remove phenol, cresol, nitrophenols, and 4-chlorophenol (*p*-chlorophenol or 4-hydroxy-chlorobenzene) from wastewater [23,29]. For the first time, the toxicity of novel polymers for different bacterial strains and the effect of freezing of cells during cryogel preparation were evaluated using spectrophotometric methods i.e., the MTT(3-(4,5-dimethylthiazol-2-yl)-2,5-diphenyltetrazolium bromide) assay [29]. It was shown that PVA-al is relatively nontoxic, whereas the toxicity of PEI-al is comparable to PEI alone [29].The toxicity of PEI derivatives is most probably related to their positive charge and the tight electrostatic adsorption on the negatively charged cells’ membrane, thus blocking some ion channels and changing the geometry of the cells. Mixing of the bacteria suspension with PEI-al solution produced an instant aggregation due to the high reactivity and concentration of aldehyde groups in the mixture. To partially neutralise the toxic effect of positively charged PEI-al on bacteria, the use of a combination of PVA-al and PEI-al with a lower concentration of PEI-al was suggested. Conditions for the preparation of the macroporous material using specific phenol-degrading bacterial strains (*Pseudomonas mendocina*, *Rhodococus koreensis*, and *Acinetobacter radioresistens* isolated from oil-contaminated soil) were optimised [23,29]. It was reported that only 10% viable cells remain after cryopolymerisation of AAm or usage of epoxy resins [96].Freeze-drying kills up to 98% of Gram-negative bacterial cells [97].

The viability of the cross-linked bacteria can be checked using Live/Dead BacLight fluorescence staining kit and Laser Scanning Confocal Microscopy (LSCM) and the metabolic activity of cells applying MTT assay [23,29]. A consortium of bacteria viability on different levels of biofilm or hydrogel can also be estimated by oxygen consumption using an O_2_ microsensor with a sensor tip in the range of 20 to 100 um [15]. BioMate luminometer Biosensor based on 2 × 10^9^ cells/mL entrapped in a14% PVA cryogel revealed more than 90% cell survival after preparation. The sensor allows phenol and chlorophenol concentrations to be detected in waste water in the range of 10 to 200 mg/L [27]. Another useful assay to evaluate the viability of immobilised cells with hydrogel structure is to estimate the concentration of intracellular adenosine triphosphate using bioluminescent luciferin–luciferase method, which concludes in dimethylsulfoxide extraction and treatment with luciferin–luciferase reagent [20].

LSCM revealed the presence of viable bacteria with the PVA-al and PEI-al cross-linkers, whereas the control material cross-linked with GA contained mostly dead cells. Bioreactors based on these bacterial-PVA-al-PEI-al cryogels were prepared for phenol degradation in batch mode at an initial concentration of 50 mg/L, pH 7.5 in carbonate buffer at a temperature of 30 °C. The utilisation of minimum salt media buffer (MSM) facilitated the degradation of higher concentrations of contaminant up to 300 mg/L. The cryogel material can be reused for at least 10 bioremediation cycles with phenol in MSM without a decline of activity [29]. Regardless of whether bacterial strains were immobilised, *Pseudomonas*, *Rhodococcus* and *Acinetobacter* spp. cryogels had an adaptation period to phenol and m-cresol of 48 h and 90 h, respectively. The first cycle of phenol 60 mg/L was consumed within 125 h, whereas the secondand third cycles of 100 mg/L phenol were degraded by cryogel unit within 90 h with efficiencies of 2.18 and 1.84 mg∙(L∙h)^−1^, respectively [29]. Various concentrations of phenol (100, 200, 300 mg/L, 200 mL) were consumed by *Acinetobacter* spp. cryogels in a static mode within 70, 145, and 170 h, respectively [29]. For comparison, more sophisticated systems used two-chamber Microbial fuel cell and open-cell bioelectrochemical systems utilising phenol 63.6 and 67 mg/(L-day), respectively [7]. The PVA–PEI cross-linker combinations exhibited good stability and viability and had the ability torecycle for at least five weeks in a carbonate buffer, utilising the contaminant as the only source of carbon. In contrast, the GA cross-linked cryogels only showed insignificant ability to degrade phenol and were non-functional after one week [23]. 

Another report ascribes a biofilm formation on the PVA-based cryogel via incubation with activated sludge for 8 days. The remediation system consisting of 2.2 L fixed-bed bioreactor inoculated with activated sludge was operated at 35 °C, oxygenation at air flow of 0.8 L/min for 95 days under a continuous flow condition for removal of nitrogen (ammonia) from wastewater. The ammonia oxidation rate was achieved up to ~1.68 kg _nitrogen_ m^−3^ d^−1^. Cryogel immobilised anammox bacteria was reactivated after oxygen shock with a short delay. The two-stage system demonstrated ~90% nitrogen removal efficiency at a short hydraulic retention time (7.7 h) [98] (Figure 5). The co-immobilization of anammox and ammonia-oxidizing strains into a PVA for efficient nitrogen removal process was proposed. Author states that it can be reused for nine consecutive, repeated batches. It was observed an equal utilization of nitrogen in both ions, NH_4_^+^ and NO_3_^−^, reaching 100% nitrogen removal during 6 cycles. The maximum of the nitrogen removal kinetic rate was equalled to 3.46 mg _nitrogen_/L∙h. The complete nitrogen removal was achieved with an initial concentration of 0.1 g _nitrogen_ L^−1^ for NH_4_^+^ and 0.06 g _nitrogen_/L for NO_3_^−^, during the 8th bioremediation cycle. [99,100]. The degradation of 2-methylphenol utilizing immobilised acclimated biomass incorporated within PVA-alginate hydrogel and cryogel beads illustrated the contaminant removal at following conditions: the optimum dry biomass density in the cryogel beads of 3–4 mm diameter was about 20 mg∙cm^−3^. As expected when additional adsorbent used this leads to enhanced performance without much biomass leakage. Thus, 0.5% (w/v) powdered activated carbon PVA-cryogel containing immobilised biomass revealed better 2-methylphenol removal and were able to sustain 9 cycles of usage with a removal efficiency of over 82% per remediation cycle (0.3 g/L of 2-methylphenol [86].

*Bacillus* sp. entrapped within hydrogel of PVA cross-linked with boric acid and sodium Alg cross-linked with calcium ions was adapted for polycyclic aromatic hydrocarbons removal. The gel preparation consists of sterile polymer solution preparation, mixing with bacteria and shaking for 8 h, followed by culturing of beads in the beef-protein medium over 48 h. Accelerated phenanthrene degradation was observed. The contraction design of the bioremediation process provided minimised oxidative damage, while superoxide dismutase (SOD) activity increased from 56 to 81 U/mg for Cd ions content in the range of 0 to 200 mg/L; however, SOD decreased to 44 U/mg at Cd ions content of 300 mg/L for the non-immobilised system, whereas the SOD kept increasing from 52 to 473 U/mg for the PVA-Alg immobilised system exposed to Cd ions in the range of 0 to 300 mg/L [101]. The same immobilisation strategy was applied for *Lactobacillus rhamnosus* ATCC7469 into PVA-Alg-Ca gel for lactic acid fermentation. Significant cell proliferation in further fermentation stages achieved was detected up to 10.7 logCFU/g. PVA-Alg-Ca cryogel with bacteria revealed better fermentation efficiency (by 37.1%) in comparison with the free cell fermentation system in batch mode. The highest lactate yield and volumetric productivity of 97.6% and 0.8 g/L∙h, respectively, were observed in repeated batch fermentation. The cryogels showed high mechanical and operational stability reaching an overall productivity of 0.78 g/L∙h over 7 consecutive batch cycles [102].

*Bacillus cereus* cells immobilised into pAAm cryogel were tested for crude oil degradation process, and the results were compared with bacteria in the planktonic state. The aliphatic hydrocarbons were degraded by 93% in the MSM over 48 h at 28°C. Reusability tests exhibited that the hydrocarbon degradation ability of immobilised microorganisms was stable over 47 days in a shaking regime; moreover, the degradation rate of immobilised cells was kept at a high level up to the 20^th^ cycle. The bacteria immobilised in cryogels exhibited more efficient degradation for 22 active cycles compared to three cycles performed by cells in planktonic state [103]. The aforementioned remediation systems at a stage of upscaling to pilot stage should be tested using several types of bioreactors design, to find the most optimal conditions for water treatment and also the lifetime of the bioreactor. An overview of various microbial bioreactors design (airlift bioreactor with external recirculation and internal recirculation; packed bed bioreactor; bio trickling filter; slurry reactor and aerobic continuously stirred tank bioreactor) have been discussed with a number of illustrations of the schematic principle of work. For instance, a 2000 m^2^ biofilter demands 160,000–360,000 m^3^/h of air. Thus, biofilters can also be applied to treat VOCs (benzene, toluene, ethylbenzene, xylenes), and the main limitation of this treatment is the inability to control the pH [5]. Advances and drawbacks are illustrated for each particular bioremediation task for a certain range of contaminants in waste water [5,104]. A recent review describes the advantages and disadvantages of remediation of toxic aromatic derivatives by using a number of oxidoreductive enzymes. There are several advantages of immobilised enzymes over the chemical treatment of water. Immobilisation on polymer particles(rodlike cellulose, Eupergit@C, Microporous polypropylene hollow fibers, cinnamic carbohydrate esters) or inorganic particles(activated alumina, aluminum-pillared interlayered clay, Fe_3_O_4_ NPs, porous celite beads) surface is a cost-effective, environmentally friendly, and highly selective process that does not generate additional wastes. Enzymatic remediation of these compounds has been explained as the transformation of phenolic compounds into insoluble products [105]. However, this approach has disadvantages such as that enzymes are easily inactivated by a harmful environment (heavy metal ions) and contamination by bacteria and fungi. Immobilised bacteria do not have the aforementioned drawbacks. It is a relatively robust and sustainable system that self-reproduces over time, and therefore, the decline of the remediation rate or purification process does not occur. Moreover, this system does not require complete replacement, but probably a short-term reactivation stage in a favourable culturing medium and temperature. *Bacillus thuringiensis* cells immobilised and entrapped in Xanthan gum hydrogel illustrated a survival rate of 99%. A negative effect of entrapment in Xanthan gum polydopamine hydrogel was registered for *Planococcus* spp. S5, illustrating viability rate in the range of 93–50% [106]. Xanthan gum polydopamine composites showed a high stability in a wide pH range, and their adsorption capacity for negatively and positively charged molecules, for instance, the adsorption capacity for Methylene blue and Congo red, were 0.578 mg/mg and 0.19 mg/mg, respectively. Naproxen bioremediation (1 mg/L in 14 days) and enhanced chemical oxygen demand removal by the entrapped *Bacillus thuringiensis* cells in Xanthan gum polydopamine composite can be considered for application for some post-water-treatment processes [106]. Prabu and Thatheyus had illustrated the efficiency of AAm degradation by planktonic and immobilised *Pseudomonas aeruginosa* via monitoring pH and ammonia cations release. The biodegraded 1–2% AAm led to the release of 7 mg NH_4_^+^ L^−1^. As expected presence of mercury and chromium ions inhibited bioremediation, however nickel(II) (at concentrations 200 and 400 ppm) accelerated the process. Planktonic *Pseudomonas aeruginosa* started active metabolic activity only after incubation over 24 h, while immobilized cells initiated biodegradation before 24 h [107]. The bioremediation of real wastewater contaminated with phenol and chlorophenol indicated better results when compared to the model systems in carbonate buffer and MSM. The above technique is more rapid and simpler than a previously published three-step cryogel surface immobilisation technique for phenol-degrading bacteria by Satchanska et al. [108]. Here, cryogel preparation was based on strong UV treatment of polyethylenoxide (PEO) and a washing step. Biofilm formation was completed after several days, and maximum phenol biodegradation was achieved after 3–4 days. A drawback is that PEO has high costs for disposal after its use. It was stated that the PEO-biofilter was able to resist and degrade phenol at a concentration of 1000 mg/L for at least 4 weeks.

There are several cases where amphoteric cryogels based on natural polymers such as CHI and gelatine, alginate and chondrotinesulphate, and heparin were developed for biomedical purposes, [35] but were quite expensive for environmental application [55,109]. A number of strategies of fabrication of composite cryogels with iron nanoparticles for efficient removal of dissolved As(III) and different medium in a flow mode were illustrated [110,111]. Synthetic amphoteric cryogels based on allylamine and sulfopropylacrylamide as well as acrylic acid were designed for the removal of heavy metal ions such cadmium and mercury. In these papers, the Medusa software was applied to model the existence of various species of Hg and Cd ions in a wide pH range, and various mechanisms of metal ion removal from water were proposed. Amphoteric cryogels illustrated adsorption capacity in the range of 132–249 mg/g and the leaching trials showed the stability of Cd2^+^ in the cryogel structure [112,113,114]. It was shown the potential of cryogels with in situ formed gold nanoparticles to be used for many environmental applications such as the conversion of toxic aromatic nitro derivatives from water and selective adsorption of mercury, via amalgam formation, which can be considered as an efficient water pretreatment stage for following bioremediation processes [44]. Two-step-prepared composite amphoteric cryogels loaded with silver nanoparticles in the range of 159 and 98 mg/g (AgNPs) were used for efficient removal of iodine anions, and it is proposed that this material has appropriate kinetics of radioactive iodine removal from aqueous solutions [115]. Contamination by bromate is commonly associated with disinfection, and therefore its removal from drinking water is crucial. Hajizadeh et al. has illustrated the preparation and application of CHI molecularly imprinted polymer (MIP) as well as composite cryogels with particles (Fe_2_O_3_·Al_2_O_3_·xH_2_O) for removal of bromate (NaBrO3) from water. The adsorption studies were carried out in the presence of nitrate. Inorganic adsorbent removed bromate quicker than the MIP cryogel, illustrating capacities of 0.8 mg/g alumosilicate composite adsorbent(Fe_2_O_3_·Al_2_O_3_·xH_2_O cryogel) and the MIP 0.2 mg/g, respectively [115]. The continuation of this research comprised the design of novel MIP cryogels for the removal of highly toxic metabolites(cisplatin, chloramphenicol) and pharmaceutical substances from water [42,116]. Thus, copolymerised methacrylic acid HEMA cryogel exhibited high affinity for cisplatin, with an adsorption capacity of 150 mg/g, which can be recycled up to 14 times [42]. Silica-grafted MIPs cryogel revealed the maximum monolayer adsorption capacities for chloramphenicol, Si@MIPs-CAP (32.26 mg∙g^−1^), which were slightly higher than nonimprinted cryogel Si@NIPs-CAP (29.6 mg∙g^−1^) [116]. Singh et al. illustrated the strategy of preparation of polymeric carriers utilising 10%, 20%, 30%, 40%, and 50% acrylamide and bacteria suspension at gamma radiation doses in the range of1 to 5 kGy, interestingly that bacteria illustrated viability, as it is well known, that some gamma radiation is used for sterilisation process. Bioaccumulation of caesium by free and immobilised bacterial cells was illustrated. Significant reductions of 76–81% caesium were noticed for bacterial cells immobilised by radiation polymerization [117]. The removal of 54% to 66% was observed applying free cells after 240 h; moreover, immobilised cells showed efficiencies of 76% to 81% bioremoval for 30% and 40% acrylamide polymer carries, respectively. Another study showed a way to reduce sulphate with bacteria immobilized into magnetic macroporous polymer beads. Magnetic particles could restrain the enrichment phenomenon of silicone and, therefore, increase the loading amount of the biofilm. At a sulphate anion concentration of 3000 mgL^−1^, the sulphate-reducing effectiveness on the magnetic support reached 40%, which was higher compared to other systems with productivities of 30%. In the long-term stage, the sulphate-reducing effectiveness of the biofilm on the magnetic poly(St-MTQ) Pickering high internal phase emulsions (HIPE) (MPSMH) and poly(St-MTQ)HIPE (PSMH) supports did not show a significant difference because of the decomposition of the ferriferous oxide [118].

## 6. Conclusions and Perspectives

In summary, for bioremediation processes, microorganisms immobilised onto a substrate show many advantages over free planktonic bacteria systems. These include higher biomass density, higher metabolic activity, and resistance to toxic chemicals. They also enable continuous operation, avoiding the requirement for biomass–liquid separation. The immobilised bacteria can be reused several times, which opens the opportunity for developing cost-effective processes for wastewater treatment.

The immobilisation of cells, including bacterial, fungal, algal, and tissue, is commonly applied in biotechnological applications at the laboratory scale and in some industrial applications (Table 1). Such systems show distinct advantages over planktonic and non-immobilised approaches, such as degradation efficiencies, cryogel stability, and cell viability and preservation. Factors influencing the adsorption or cross-linking of cells onto the cryogel matrices include their surface structures, cell surface charge, pore size, and hydrophobicity [119,120]. The efficacy of attachment is further compounded by the medium used, the pH and temperature of the surrounding milieu, the presence of other contaminants and chemicals, and the properties of the absorbents exploited [98,121].

The specific properties of macroporous cryogels, including their biocompatibility, physical and chemical stability and strength, and controlled pore structure, make them potentially important tools in the remediation and processing of various chemical and biological components and contaminants. Their controlled and large pore size enables the rapid processing of liquids for cell separation and chromatography, filtration and the specific capture and detection of pathogens, and for the bioremediation of chemical contaminants in water.

However, specific differences in applications, processes, and goals preclude a standardised and universal method of cell immobilisation. Aldehyde-based polymers revealed efficient cross-linking, but at the same time most derivatives possessed high toxicity. There is a need to develop a novel, functional, low-toxicity, and cheap polymer for efficient cross-linking of cells and with appropriate mechanical properties. The pre-freezing method is a simple and efficient means of controlling pore size and particle distribution, leading to opportunities to develop novel composite macroporous materials with large pore size and surface area, offering enhanced adsorption and separation characteristics. Together with other advances in our understanding of cryogel chemistry and synthesis, there exists the possibility of Tabkledesigning different micro- and macroporous, multicomponent structures of different sizes and porosities for numerous applications in the biotechnological, environmental, and medical fields.

## Figures and Tables

**Figure 1 polymers-13-01073-f001:**
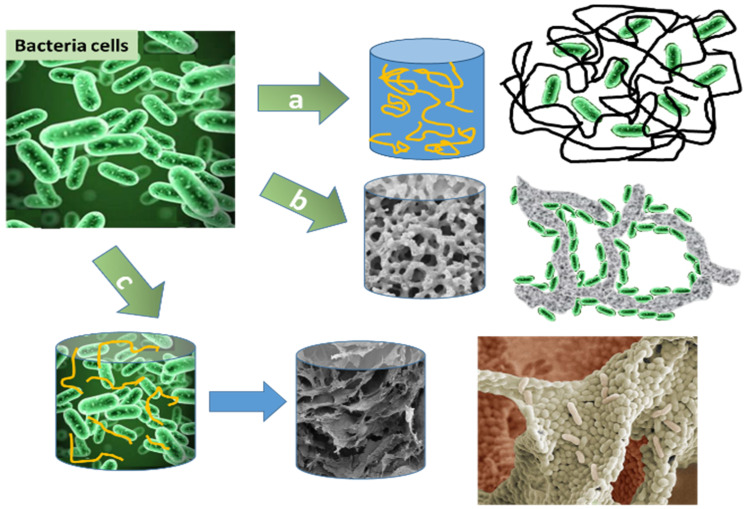
Strategies for bacterial cell immobilisation onto scaffold/polymer supports, and their benefits and limitations: (**a**) physical entrapment of bacteria cells within hydrogel structure; (**b**) biofilm generated on the porous surface of inorganic support; (**c**) macroporous gel formation via direct covalent linkage of bacteria cells at cryostructutation process [23].

**Figure 2 polymers-13-01073-f002:**
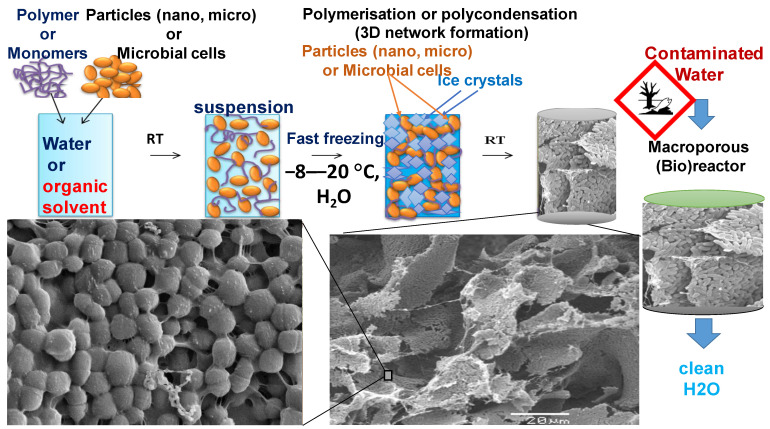
Scheme of preparation of cryogels from bacterial cell suspension and cross-linking polymer(left to right), mixing of components, freezing of the bacteria suspension and thawing step; illustration of the morphology of macroporous structure(SEM microphotograph at low and high magnification). The left part illustrates the principle of the water bioremediation process by macroporous bioreactor in flow-through mode.

**Figure 3 polymers-13-01073-f003:**
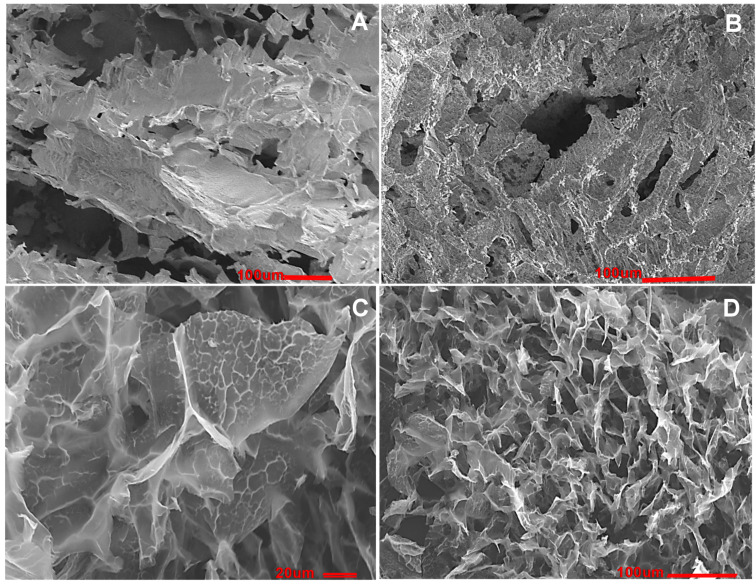
SEM microphotographs of composite cryogels: (**A**,**B**) composites PVA-CHI-hydroxyapatite-heparin-GA; scale bar 100; (**C**) polyelectrolyte complex PVA-CHI-heparin-GA scale bar 20 [35] and (**D**) CHI-GA; scale bar 100 [44].

**Figure 4 polymers-13-01073-f004:**
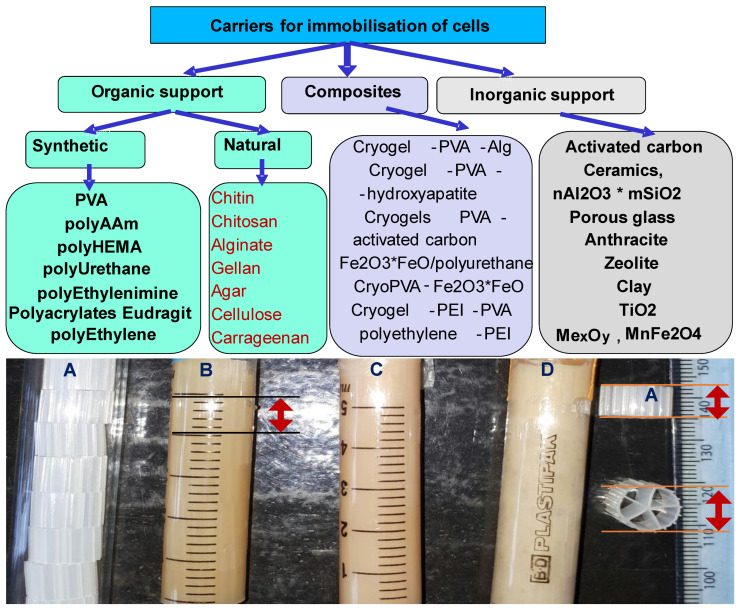
Immobilization carriers: (**A**) polypropylene polymeric Kaldnes carriers; (**B**)composite cryogels prepared in Kaldnes carriers PVA-al-PEI-al-*Pseudomonas* spp. (**C**) composite cryogels prepared in Kaldnes carrier PVA-al-PEI-al-*Athrobacter Chlorophrnolicus* and (**D**) composite cryogels prepared in Kaldnes carrier PVA-al-PEI-al-*Acinetobacter* spp.

**Figure 5 polymers-13-01073-f005:**
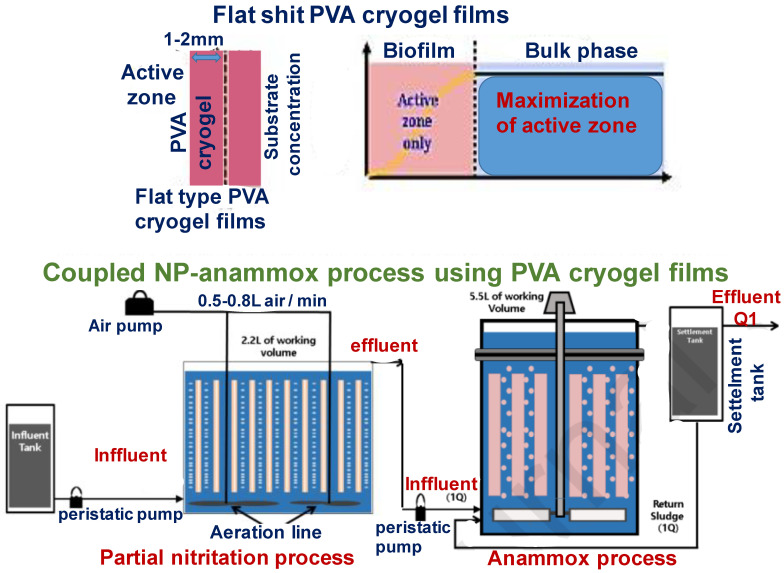
Experimental setups for continuous flow fixed-bed bioreactors (FBBRs) with flat-type PVA cryogel film [98] adapted with permission from Elsevier.

**Table 1 polymers-13-01073-t001:** Comparative analysis of polymer used for immobilization of cells and its application.

Microorganism	Polymer Type or Cross-Linker	Application	Reference
*Aspergillus awamori*	polyacrylonitrile membrane	phenol biodegradation	[21]
*Clostridium acetobutylicum*, *E. coli*, *Pseudomonas* spp., *Rhodococcus* spp.	glutaraldehyde	biofuel, bioremediation	[23,62,78]
*E. coli*, *Clostridium acetobutylicum*, *Pseudomonas* spp., *Rhodococcus* spp., *Acinetobacter* spp.	PVA-aldehyde/PEI aldehyde/oxidized dextran or aldehyde dextran	hydrolysis/fermentation/bioremediation	[23,28,29,62]
*Actinobacillus succinogenes*, *Rhizopusoryzae* spp.	PVA-cryogel	lactic, fumaric and succinic acids	[51]
*Chlorella* spp.	Alg, Carrageenan, Agarose, Alginate and Agar beads, polyurethane foam	removal of ions Ni, Zn, Cd, Cu, Hg, Pb, and uranium, phosphate, nitrite, NH4	[52,111,112,113]
*Pseudomonas citronellolis*	PVA bamboo-biochar beads	toluene and hydrocarbons	[46]
*Trichoderma* spp.	HEMA cryogel	cyanide removal	[55]
*Komagataei bacterxylinum*	PVA-cryogel	microcellulose	[58]
-	polylysine-b-polyvaline GA cryogel Gum Arabic linked with divinyl sulfone	water treatment, antimicrobial activity E-coli	[54]
-	aldehyde modified dextran	scaffolds or mammalian cell immobilisation	[28,82]
*Pseudomonas fluorescens(S3X)*, *Microbacterium oxydans (EC29)*	hydroxyapatite	removal of zinc and cadmium ions	[68]
*Pseudomonas rhodesiae*, *Bacillus subtilis*, *Bacillus lateroporus*	cryogel polyethylenoxide UV	phenol, methylphenol/cresol remediation	[85,86,122]
Nitrosomonas europaea C-31 and ‘Candidatus *Jettenia caeni*, *Rhodococcus* spp., *Bacillus* sp., *Pseudomonas* spp.	PVA cryogel	removal of ammonia	[90,98]
*Bacillus*	PVA- H3BO3-Ca- Alg beads	polycyclic aromatic hydrocarbons removal	[101]
*Lactobacillus rhamnosus*	PVA-Alg-Ca cryogel	lactate production	[103]
*Bacillus cereus*	pAAm-BisAAm cryogel	crude oil degradation	[104]

## Data Availability

Not applicable.

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
