# Peer review of "Polymeric Materials Used for Immobilisation of Bacteria for the Bioremediation of Contaminants in Water"

_polymers, 2021, doi:10.3390/polym13071073_

Round 1
Reviewer 1 Report
There mainly grammatical errors
Line 36. Some processes require high temperatures and extreme pressures (e.g. hydrogenation, hydrodechlorination), and/or expensive catalysts such as platinum, palladium, rhodium and gold nanoparticles oncarbon or other supports [1-5]. are not always cost effective.
CORRECT: Some processes require high temperatures and extreme pressures (e.g. hydrogenation, hydrodechlorination), and/or expensive catalysts such as platinum, palladium, rhodium, and gold nanoparticles on carbon or other supports [1-5], and are not always cost-effective.
Line 40 For example, the Fenton oxidation process may result in production of polychlorinated dibenzo-p-dioxins and dibenzofurans [3,4,6] and chemical oxidation using iron nanoparticles for the dechlorination process [5]. REMARK: Rewrite the sentence explaining effects of oxidation using iron nanoparticles
Line 42 Several book chapters are devoted to immobilisation strategies for microorganisms (prokaryotes and eukaryotes, primarily fungi) on various organic (bark or wood chips) and inorganic supports (Fe2O3, TiO2, Al2O3x SiO2, 44 cement particles) are presented [7,5]. CORRECT: Several book chapters are devoted to immobilisation strategies for microorganisms (prokaryotes and eukaryotes, primarily fungi) on various organic (bark or wood chips) and inorganic supports (Fe2O3, TiO2, Al2O3x SiO2, cement particles) [7,5].
Line 48: Advantages and drawbacks of each immobilisation approach (artificially generated biofilm, biofilm produced in nature) compare to microorganisms in a plank-49 tonic state for treatment of waste water from oil, herbicides, pesticides, xenobiotic and heavy metals is discussed [7]. CORRECT: Advantages and drawbacks of each immobilisation approach (artificially generated biofilm, biofilm produced in nature) compare to microorganisms in a planktonic state for purifying of waste water from oil, herbicides, pesticides, xenobiotics and heavy metals are discussed.
Line 65: poly-acrylamide, polyvinyl alcohol (PVA), polyethylene-glycol (PEG) and polycarbamoyl sulphonate, polypropylene, polyethylene, polyvinylchloride, poly-urethane, polyacrylonitrile CORRECT: polyacrylamide (PAM) , polyvinyl alcohol (PVA), polyethylene glycol (PEG) and poly(carbamoyl sulphonate) (PES), polypropylene (PP), polyethylene (PE), polyvinylchloride (PVC), polyurethane (PU), polyacrylonitrile (PAN)
Line77
The management of bioremediation process parameters such as oxygen concentration, pH, sulphide and nitrite/nitrate levels are important especially processes utilising entrapped bacterial cells.
CORRECT: The management of bioremediation process parameters such as oxygen concentration, pH, sulphide and nitrite/nitrate levels is important especially for processes utilising entrapped bacterial cells.
Line 111
Biofilm develops in following steps motile cells attach to the surface, maturate and produce protective extracellular matrix and finally biofilm dispersal [20]. CORRECT: Biofilm develops in following steps: motile cells attach to the surface, maturate and produce protective extracellular matrix and finally biofilm dispersal [20].
Line 115 This filters proposed sample preparation CLARIFY: No filters were mentioned above
Line 124 (diffusion low density of cells relatively hydrated polymer volume ratio) PLEASE MAKE IT CLEAR
Line 127 Microorganism CORRECT Microorganisms
Line 128
due to cells are incorporated in biofilm matrix, acting as a physical barrier CORRECT: due to cells incorporated in biofilm matrix, acting as a physical barrier
Line 131
Macroporous gel formation via direct covalent linkage of bacteria cell’s membrane into 3D biofilm(one step, rapid biofilm formation due to cryoconcentration 132 phenomenon leading to generation of high cells density with respect to polymer weight) [21,22]. COMMENT: There is no verb in this sentence
Line 140 closed fuel cell system composed the anode and cathode CORRECT: closed fuel cell system composed of the anode and cathode
Line 144: containing of anode and cathode CORRECT: containing anode and cathode
Line 145 resulting protons is reduced to hydrogen gas that is supplied by an external voltage. COMMENT: gas is supplied by an external voltage? BETTER: resulting protons are reduced to hydrogen atom, forming gaseous hydrogen molecules.
Line 147 formation of CH4 BETTER formation of methane
Line 149 bio catalytic CORRECT biocatalytic
Line 161 Cryogels possessing a spongy 3D polymeric network structure synthesized via a cryogelation
CORRECT Cryogels possessing a spongy 3D polymeric network structure are synthesized via a cryogelation
Line 169 Other approaches include irradiation of the polymer solution or suspension with gamma-rays (electron beam, UV irradiation) [39] or modified polymers and self-assembly of supramolecular gelators CORRECT 169 Other approaches include irradiation of the polymer solution or suspension ( gamma-rays (electron beam, UV irradiation) [39] or modification polymers and self-assembly of supramolecular gelators
Line 207 Polymers containing amino- hydroxyl- thiol- can be functionalised CORRECT Polymers containing amino- hydroxyl or – thiol groups can be functionalized
Line 217 Hydromymethylmethacrylate CORRECT hydroxymethylmethacrylate
Lines 232, 576: bio trickling CORRECT biotrickling
Line 258 Lysozyme imprinted bacterial cellulose (Lyz-MIP/BC) nanofibers through was developed CORRECT Lysozyme imprinted bacterial cellulose (Lyz-MIP/BC) nanofibers was developed
Lines 287, 291, 294, 300 : plastic core BETTER polymer core
Line 294 – Figure 4
Alginates are not synthetic polymers, but natural
REMARK: Please use accepted abbreviations for synthetic polymers
polyethyleneimin
Lines 321, 598
flow through CORRECT flow-through
Line 297 removal of E2 QUESTION: what is E2?
Line 398: E2-MIP/MGPs QUESTION: What is MGPs
Line 421 cleavage predominantly occur CORRECT cleavage predominantly occurs
Line 565
Bacillus cereus cells immobilised into pAAm cryogel were tested for crude oil degradation process and compared results with bacteria in planktonic state. CORRECT: Bacillus cereus cells immobilised into pAAm cryogel were tested for crude oil degradation process and THE results WERE compared with bacteria in planktonic state.
Line 585
Immobilised on polymer particles(rodlike cellulose, Eupergit@C, Microporous polypropylene hollow fibers, cinnamic carbohydrate esters,) or inorganic particles(activated alumina, aluminum-pillared interlayered clay, Fe3O4 NPs, porous celite beads,) surface and its usage is cost-effective, environmentally friendly, and highly selective process, that does not generate additional wastes. CORRECT Immobilisation on polymer particles(rodlike cellulose, Eupergit@C, Microporous polypropylene hollow fibers, cinnamic carbohydrate esters,) or inorganic particles(activated alumina, aluminum-pillared interlayered clay, Fe3O4 NPs, porous celite beads,) surface, and its usage is a cost-effective, environmentally friendly, and highly selective process, that does not generate additional wastes
Line 589
Enzymatic remediation of these compounds has explained as the transformation of phenolic compounds into in-590 soluble products CORRECT Enzymatic remediation of these compounds has been explained as the transformation of phenolic compounds into in-590 soluble products
Line 594
that self-reproduce over time and therefore decline of REMARK: Please explain what do you mean by “ and therefore decline of”
Line 608
kinetic of radioactive iodine removal CORRECT kinetics of radioactive iodine removal
Author Response
Dear editor,
We are so grateful for the comments and feedback of receivers, who improved quality of the manuscript.
we agree with numerous typo, that were corrected in the text. All typographical mistakes were corrected within the text.
Comment Line 124 (diffusion low density of cells relatively hydrated polymer volume ratio) PLEASE MAKE IT CLEAR
Response "restricted diffusion of substrate to cells and corresponding metabolism products, low density of cells relatively hydrated polymer volume ratio, high viability of cells"
Comment Line 297 removal of E2 QUESTION: what is E2?
Response E2 is abbreviation for 17beta-estradiol
Comment Line 594that self-reproduce over time and therefore decline of REMARK: Please explain what do you mean by “ and therefore decline of”
Response It is relatively robust and sustainable system, that self-reproduce over time and therefore decline of remediation rate or purification process does not occur. Moreover, this system does not require complete replacement, but probably a short term reactivation stage in a favourable culturing medium and temperature.
Best regards,
Dmitriy Berillo
Reviewer 2 Report
It is an interesting review paper. I would consider some minor revision to improve the impact but the MS itself is almost ready for publication.
- a careful revision of the conclusion section is suggested. Indeed, the authors summarize the content of the review, but they fail to underline the critical findings reported herein, which constitute the true essence of a review article.
- A review should give basic information to beginners and comprehensive information to experienced researchers in this field. However, this review needs a little more of background information of the biological aspects that might be missed from the readers of Polymers journal.
- The figures need revision. Fig 3: labels for scale bars are not clear. Fig 4 does not provide a clear message I had to read the text twice to understand it.
Author Response
Authors thankful to reviewer for the valuable comments.
We did our best to improve the quality and address all comments. Some parts of the review were paraphrased and additional information added.
Comment Fig 3: labels for scale bars are not clear.
Answer the Scale bars are included in the capture of the figure 3.
Comment Fig 4 does not provide a clear message I had to read the text twice to understand it.
Answer Figure 4 Immobilization carriers: A) polypropylene polymeric Kaldnes carriers; B)composite cryogels prepared in Kaldnes carriers PVA-al-PEI-al-Pseudomonas sp. C) composite cryogels prepared in Kaldnes carrier PVA-al-PEI-al-Athrobacter Chlorophrnolicus and D) composite cryogels prepared in Kaldnes carrier PVA-al-PEI-al-Acinetobacter sp..
Reviewer 3 Report
The proposed manuscript is a comprehensive review on Polymeric materials used for immobilisation of bacteria for the bioremediation of contaminants in water, focusing on production, types, main characteristics and applications of macroporous materials, including cryogels.
I consider this material can be published with several modifications, as suggested below:
Authors should use either Figure, or Fig. in the text when referring to their Figures, not both.
Page 1, lines 38-39: Dot should be deleted and “therefore” should be introduced after [] in “..or other supports [1-5]. are not always cost effective”
Page 2, line 71: Figure (number) is missing? The phrase has no meaning.
line 72: Please mention full name for GA when first using it.
Page 3, line 114: Replace WAS with WERE when referring to cryogels.
lines 140-142: This phrase should be modified for a better understanding.
line 145: Replace IS with ARE in “resulting protons is reduced”.
Page 4, lines 161-162: Verb is missing in this phrase.
Page 5, line 224: Italics should be used for Chlorella sp.
Page 6, line 240: Italics should be used for Trichoderma sp.
line 254: R is missing in “bioreacto”
line 258: What is “through” referring to in this phrase?
Page 7: Figure 2 should be modified as there are 2 images repeating in this figure, while the image left down more than 50% similar with one image from Figure 1.
The title of Figure 2 should be modified as well for a proper understanding.
Page 9: Figure 4 – please mention differences for the 3 syringes containing composite cryogels. The whole Figure 4 should be diminished to proper fit the page.
Fe3O4, nAl2O3 x mSiO2, TiO2, MnFe2O4 should be correctly written.
Page 11, line 389: Which is the reference for Bo et al.?
Page 15: Parts of Figure 5 cannot be clearly observed.
lines 553-555: Please mention the full name for SOD abbreviation.
Author Response
Authors thanks reviewer for the valuable comments.
All typographical errors were corrected within the text using option track of changes.
comment : Authors should use either Figure, or Fig. in the text when referring to their Figures, not both.
Answer: We agree with the remark and corrected to "Figure"
comment : Page 2, line 71: Figure (number) is missing? The phrase has no meaning.
Answer:The sentence was paraphrased" Bacteria have been immobilised on a range of chemically activated and/or inert supports."
The cryogelation techniques available enable the production of elastic macroporous cryogels with a wide range of porosities and morphologies for instance on figure 3 is exhibited morphology of composite cryogels based on PVA & chitosan & glutaraldehyde(GA) and PVA & CHI & hydroxyapatite & GA, respectively.
comment : line 72: Please mention full name for GA when first using it.
Answer: GA glutaraldehyde was incorporated into text at first mention.
comment : lines 140-142: This phrase should be modified for a better understanding.
Answer: Microbial electrolysis cell (MEC) is a bioelectrochemical systems containing anode and cathode, where anodic microorganisms oxidize organic derivative and the resulting protons are reduced to hydrogen atom forming gaseous hydrogen molecules. The reaction on the cathode electrode may also result in formation of methane, which depends on immobilised bacteria [7].
comment : Page 4, lines 161-162: Verb is missing in this phrase.
Answer: Cryogels possessing a spongy 3D polymeric network structure are synthesized via a cryogelation or cryostructuration technique[32, 34, 37, 44, 47]; the concept of cryogel preparation based on cells is summarised in figure 2.
comment : line 258: What is “through” referring to in this phrase?
Answer: The word "through" was removed from the sentence.
comment : The title of Figure 2 should be modified as well for a proper understanding
Answer:
Figure 2 Scheme of preparation of cryogels from bacterial cell suspension and cross-linking polymer(left to right), mixing of components, freezing of the bacteria suspension and thawing step ; illustration of the morphology of macroporous structure(SEM microphotograph at low and high magnification). Left part illustrates the principle of water bioremediation process by macroporous bioreactor in a flow through mode.
comment : Page 9: Figure 4 – please mention differences for the 3 syringes containing composite cryogels. The whole Figure 4 should be diminished to proper fit the page.
Answer: Figure 4 Immobilization carriers: A) polypropylene polymeric Kaldnes carriers; B)composite cryogels prepared in Kaldnes carriers PVA-al-PEI-al-Pseudomonas sp. C) composite cryogels prepared in Kaldnes carrier PVA-al-PEI-al-Athrobacter Chlorophrnolicus and D) composite cryogels prepared in Kaldnes carrier PVA-al-PEI-al-Acinetobacter sp..
Comment:Fe3O4, nAl2O3 x mSiO2, TiO2, MnFe2O4 should be correctly written
Answer: All structures are written correctly. May be the reviewer is not familiar with the representation of mixed iron oxide Fe3O4 which also can be written as FeOx Fe2O3
Comment: Page 11, line 389: Which is the reference for Bo et al.?
Answer:We agree with the comment. It was the name of the researcher. It was substituted to Matiasson
Comment: Page 15: Parts of Figure 5 cannot be clearly observed.
Answer: The figure 5 was prepared at higher resolution.
Comment: lines 553-555: Please mention the full name for SOD abbreviation.
Answer: SOD is superoxide dismutase , this was included to manuscript.
Best regards,
Berillo Dmitriy